# Genetic enhancers of partial PLK1 inhibition reveal hypersensitivity to kinetochore perturbations

Karine Normandin[1], Jasmin Coulombe-Huntington[1,¤a], Corinne St-Denis[1], Alexandre Bernard[1], Mohammed Bourouh[1], Thierry Bertomeu[1], Mike Tyers[1,2,¤b], Vincent Archambault[1,3]*

1 Institute for Research in Immunology and Cancer, Université de Montréal, Montréal, Canada,
2 Département de médecine, Université de Montréal, Montréal, Canada, 3 Département de biochimie et médecine moléculaire, Université de Montréal, Montréal, Canada

¤a Current address: Department of Bioengineering, McGill University, Montréal, Canada
¤b Current address: Hospital for Sick Children, University of Toronto, Toronto, Canada
* vincent.archambault.1@umontreal.ca

**Data Availability Statement:** All relevant data are within the manuscript and its Supporting Information files.

## Abstract

Polo-like kinase 1 (PLK1) is a serine/threonine kinase required for mitosis and cytokinesis. As cancer cells are often hypersensitive to partial PLK1 inactivation, chemical inhibitors of PLK1 have been developed and tested in clinical trials. However, these small molecule inhibitors alone are not completely effective. PLK1 promotes numerous molecular and cellular events in the cell division cycle and it is unclear which of these events most crucially depend on PLK1 activity. We used a CRISPR-based genome-wide screening strategy to identify genes whose inactivation enhances cell proliferation defects upon partial chemical inhibition of PLK1. Genes identified encode proteins that are functionally linked to PLK1 in multiple ways, most notably factors that promote centromere and kinetochore function. Loss of the kinesin KIF18A or the outer kinetochore protein SKA1 in PLK1-compromised cells resulted in mitotic defects, activation of the spindle assembly checkpoint and nuclear reassembly defects. We also show that PLK1-dependent CENP-A loading at centromeres is extremely sensitive to partial PLK1 inhibition. Our results suggest that partial inhibition of PLK1 compromises the integrity and function of the centromere/kinetochore complex, rendering cells hypersensitive to different kinetochore perturbations. We propose that KIF18A is a promising target for combinatorial therapies with PLK1 inhibitors.

## Author summary

Cell division requires protein kinases that add phosphates to target proteins to modify their activities. PLK1 is an essential kinase controlling multiple events in cell division. Over 3 decades after its discovery, which cellular functions are most dependent on PLK1 is unknown. Moreover, while drugs have been developed against PLK1 for cancer treatment, these compounds alone are largely ineffective. In principle, if cells become dependent on the functions of particular proteins when PLK1 activity is decreased, inhibiting these

**Funding:** This work was supported by a Project Grant from the Canadian Institutes of Health Research to VA (CIHR, PJT-152915), an Operating Grant from the Cancer Research Society to VA, a Discovery Grant from the Natural Sciences and Engineering Research Council of Canada to VA, and by a CIHR Foundation grant to MT (FDN-167277). The funders had no role in study design, data collection and analysis, decision to publish, or preparation of the manuscript.

**Competing interests:** The authors have declared that no competing interests exist.

proteins and PLK1 simultaneously might more effectively target cancer cells. To identify such potential vulnerabilities, we used a CRISPR-based screening strategy to search for genes that become essential for cell division when PLK1 activity is decreased. Most genes we identified encode proteins that function in the building and function of kinetochores, the structures by which chromosomes attach to the spindle that separates chromosomes in mitosis. Two proteins, KIF18A and SKA1, become particularly crucial for mitosis when PLK1 activity is compromised. Our results indicate that, among all cellular functions controlled by PLK1, kinetochore function is most sensitive to PLK1 inactivation. KIF18A is a druggable enzyme that may synergize with PLK1 inhibition in cancer treatments.

## Introduction

The eukaryotic cell division cycle is controlled by intricate and concerted molecular mechanisms that are evolutionarily conserved [1]. The isolation of mutants in invertebrate model organisms including yeast and flies has identified most of the essential factors that drive cell cycle progression. Moreover, the dissection of the functional relationships between these myriad factors has been facilitated by genetic screens for enhancers and suppressors followed by cellular, molecular and biochemical characterization. In most cases, subsequent studies in mammalian cells have confirmed conservation of mechanisms identified in model organisms. However, systematic genetic screens directly in human cells have not been possible until the recent advent of genome-wide CRISPR-based screens [2]. Aspects of cell division that may be uniquely critical in human cells thus remain to be identified.

Polo-like kinase 1 (PLK1) is a serine/threonine kinase that is conserved from yeast to humans [3,4]. The initial *polo* mutants isolated in *Drosophila* revealed that Polo kinase (PLK1 in humans) is required for bipolar spindle assembly and centrosome maturation in early mitosis [5]. Similar results were obtained by injection of neutralizing PLK1 antibodies in human cells [6]. The requirement for Polo function in cytokinesis was first recognized in the fission yeast *S. pombe* [7]. Selective inhibitors of PLK1 subsequently confirmed that its kinase activity is needed for cytokinesis in human cells [8–10]. Complete inhibition of PLK1 in human cells blocks mitotic entry [11]. However, studies using less acute inactivation of PLK1 or cell cycle synchronization revealed functions of PLK1 in various other mitotic processes including chromosome condensation, chromosome attachment and alignment on the spindle, removal of sister chromatid cohesion before anaphase, mitotic exit and centromere assembly [3,12,13]. Consistent with these multiple functions, PLK1 localizes to centrosomes, kinetochores (KTs) and the cytokinetic midbody [3]. These discrete localizations of PLK1 are largely mediated by its C-terminal Polo-Box Domain, which comprises a binding site for specific phosphorylated motifs [14–17]. Several cellular and molecular events in mitosis and cytokinesis have been attributed to the phosphorylation of specific effector proteins by PLK1 [3,12,13]. Despite these numerous studies, the functions that most crucially require PLK1 activity in mitosis remain unclear.

PLK1 is a validated drug target for cancer treatment [18–21]. As cancer cells often overexpress PLK1 and are hypersensitive to PLK1 inhibition [22–24], PLK1 represents a cancer-cell specific vulnerability. Several chemical inhibitors of PLK1 have been tested in clinical trials against a variety of solid cancers and leukemias [25]. While significant efficacy was obtained in several cases, complete remission was rarely achieved and therapeutic doses of PLK1 inhibitors were limited by toxicity, which manifests in various tissues including the hematopoietic system [25]. One potential strategy to overcome these limitations is co-treatment with other drugs

that specifically sensitize cancer cells to partial PLK1 inhibition. This approach has been tried in an ad hoc manner with standard chemotherapeutic drugs such as microtubule (MT) poisons and DNA replication inhibitors, as well as with various newer rationally targeted drugs, with some successes [26]. However, to our knowledge, the search for co-targets that sensitize cells to PLK1 inhibition has not been carried out systematically in a genome-wide manner.

In this study, we have used genome-wide CRISPR-based screens to identify genes that become essential when human cells are treated with low concentrations of PLK1 inhibitors. We found that inactivation of factors that promote chromosome attachment and alignment on the mitotic spindle potently sensitizes cells to a decrease in PLK1 activity. PLK1 inhibition phenotypes in these sensitized backgrounds include mitotic arrest, reduced proliferation and nuclear reassembly defects. These results suggest that the essential function of PLK1 with the highest requirement for PLK1 activity promotes chromosome attachment and segregation in mitosis. Our findings suggest potential new targets for combination therapy with existing PLK1 inhibitors.

## Results

### A chemogenomic screen for enhancers of partial PLK1 inhibition

To identify genes that become essential for cell proliferation when PLK1 is partially inhibited, we screened a clonal derivative of the NALM-6 pre-B lymphocytic leukemia suspension cell line that bears a doxycycline-inducible Cas9 expression cassette (Fig 1A) [27]. Cells were infected with a previously described custom extended-knockout (EKO) library of lentiviral vectors containing 278,754 single-guide RNA (sgRNA) sequences that cover most of the human genome at approximately 10 sgRNAs per gene [27]. Following Cas9 induction to trigger double strand DNA cleavage and error-prone repair, cells were treated for 8 days with different concentrations of PLK1 inhibitors corresponding to $IC_{10}$, $IC_{30}$ or $IC_{50}$ values, as estimated from small-scale proliferation assays in 384 well plate format (see Materials & Methods). Genomic DNA was extracted from each sample and sgRNA frequencies from the pooled libraries determined by next generation sequencing. Statistical depletion or enrichment of sgRNA sequences in control versus PLK1 inhibitor-treated pools was assessed using a version of the RANKS algorithm which controls for the effect of growth inhibition on the frequencies of essential gene targeting guides (see methods). Genes for which sgRNAs were depleted represented candidate synthetic lethal interactors (enhancers) of PLK1 partial loss of function, whereas genes for which sgRNAs were enriched represented rescues (suppressors). We repeated screens using three different PLK1 inhibitors (9 screens total): BI2536 [28], its analog BI6727/Volasertib [29], and the structurally unrelated compound GSK461364A [30].

Screens conducted with higher inhibitor concentrations that caused strong growth inhibition favored recovery of rescue clones that outgrew the rest of the pool, whereas screens at lower inhibitor concentrations favored the detection of enhancers that were depleted from the pool. We note that the estimated IC values deviated to some extent from the actual IC values observed in each screen due to culture scale up and longer duration of the screens (S1 Table). Two screens with too little growth inhibition were uninformative (i.e., 1 nM BI2536 and 1 nM BI6727 yielded no hits). Here, we focus on the results of three screens that yielded the maximum numbers of enhancers with each of the three inhibitors tested.

### Cells compromised for PLK1 activity are hypersensitive to perturbations of centromere and kinetochore functions

Screens with the three PLK1 inhibitors yielded highly similar signatures for specific cellular functions. At the gene level, pairwise comparisons of the scores obtained in screens with the

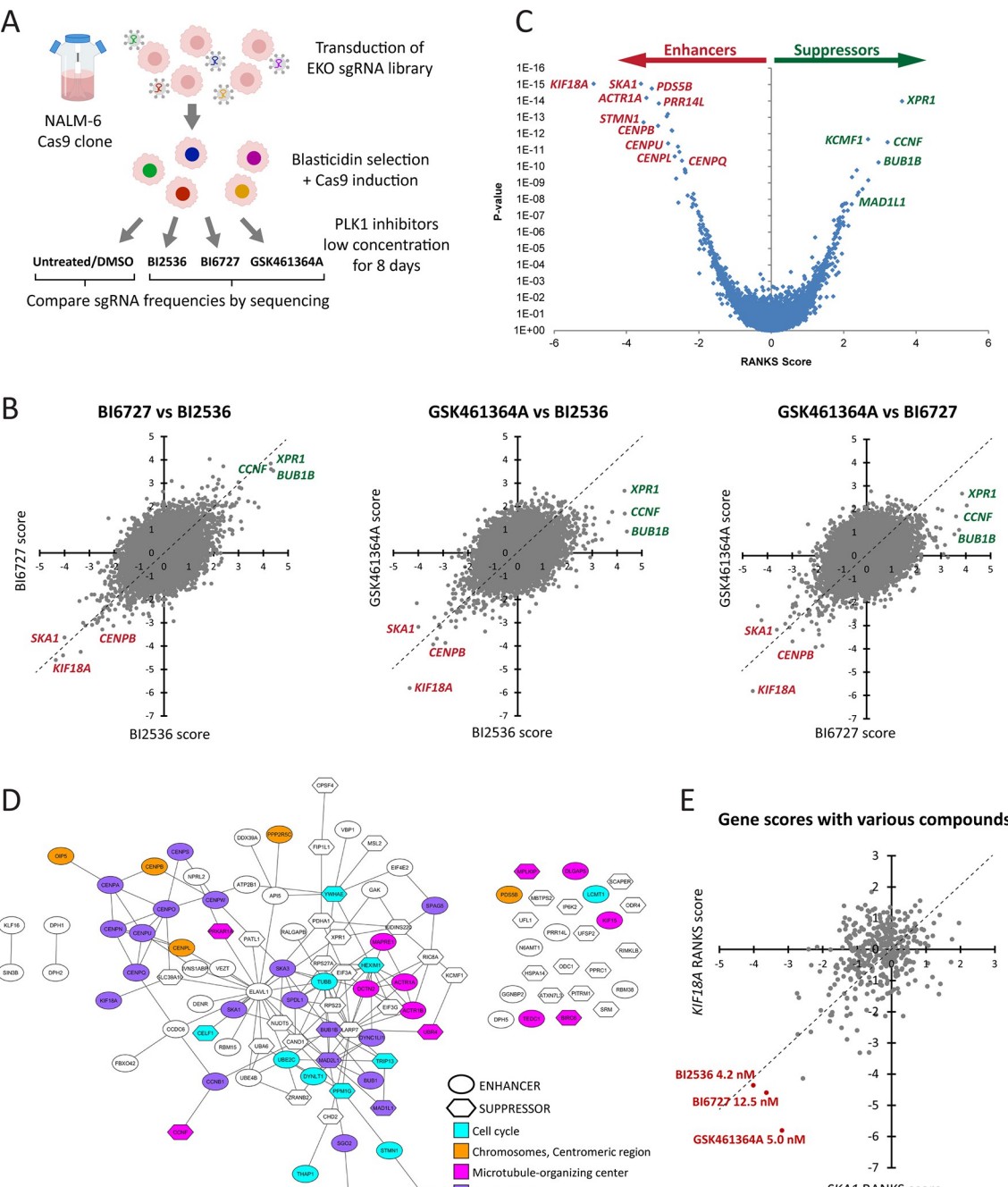

**Fig 1. A chemogenomic screen identifies enhancers and suppressors of partial PLK1 inhibition. A.** Screen design. A NALM-6 clonal cell line allowing doxycycline-inducible Cas9 expression was transduced with the EKO sgRNA library. After blasticidin selection for the presence of sgRNA constructs, doxycycline was added and cells were allowed to proliferate for 8 days with or without the compounds as indicated. Cells were then harvested and submitted to Next Generation Sequencing (NGS) to quantify the relative amounts of each sgRNA constructs in the different cell populations. **B.** Pairwise comparisons of gene RANKS scores obtained in the 3 screens with the different PLK1 inhibitors. Negative scores indicate enhancers and positive scores indicate suppressors. Names of selected hit genes of interest are labeled. **C.** Results distribution showing average RANKS score and combined P-value across the 3 screens for each gene. Names of selected hit genes of interest are labeled. **D.** Protein interaction network of genes identified as enhancers and suppressors of partial PLK1 inhibition in the screens. Interactions (lines) are those reported in BioGRID or IntACT. Only genes with False Discovery Rates (FDR) < 0.001 were included. Colors indicate selected representative enriched GO terms. FDR-corrected P-values for selected GO terms are: $6.0 \times 10^{-22}$ for Chromosomes, Centromeric region; $2.6 \times 10^{-17}$ for Kinetochore; $9.0 \times 10^{-6}$ for Microtubule-organizing center; $7.4 \times 10^{-8}$ for Cell cycle process; $5.0 \times 10^{-5}$ for Regulation of cell cycle (these last 2 terms were grouped into Cell cycle in the figure). **E.** Distribution of RANKS scores for *KIF18A* and *SKA1* in 399 screens with 303 bioactive compounds. The top 3 screens with the most extreme scores are indicated. Note that they were conducted with the 3 PLK1 inhibitors tested. Coordinate values used to generate graphs in panels B-C and E are available in S2 Table and S1 Data, respectively.

three inhibitors revealed highly reproducible enhancers and suppressors (Fig 1B and S2 Table). The GSK461364A screen was less effective in identifying suppressors, consistent with the weaker effect on cell proliferation compared to the BI2536 and BI6727 screens (S1 Table). Nevertheless, combining the results of the three screens yielded a robust PLK1-dependent signature (Fig 1C and S2 Table).

Many enhancer genes identified encoded proteins with known KT or centromere functions: *KIF18A*, *SKA1*, *CENPB*, *KIF15*, *CENPU*, *CENPL*, *CENPQ*, *CCNB1*, *CENPO*, *CENPW*, *CENPM*, *BUB1*, *CENPA* and *CENPS* were all in the top 50 enhancers (Fig 1B–1D and S2 Table). These results suggested that when PLK1 function is compromised, optimal KT/centromere functions become crucial for cell proliferation and that this genetic dependency on PLK1 kinase function dominates other known roles of PLK1. *KIF18A* and *SKA1* were the two highest-scoring enhancer genes identified. To assess specificity for PLK1 inhibition, we compared the scores obtained for *KIF18A* and *SKA1* in the PLK1 inhibitor screens against anonymized scores drawn from 399 other screens against a diverse set of 303 bioactive compounds. Strikingly, *KIF18A* and *SKA1* yielded the highest enhancer scores with the PLK1 inhibitors versus all other compounds (Fig 1E). These results suggest that the dependency of PLK1-inhibited cells on *KIF18A* and *SKA1* is highly specific and reflects a close functional relationship between these genes/proteins.

Beyond the dominant KT/centromere signature, other genetic interactions with PLK1 inhibition included cell cycle regulation and MT organization. Proteins encoded by these genes formed an extensive and statistically significant protein interaction network (Fig 1D, see methods). To assess the reproducibility of the PLK1 genetic dependencies identified in our screens, we selected 12 genes from this network for validation. Two sgRNAs from the EKO library were chosen for each gene, based on high scores in the primary screen. As negative controls, we used sgRNAs targeting the functionally inert Adeno-Associated Virus integration Site 1 (*AAVS1*) locus and the irrelevant Azami-green sequence. Following lentiviral transduction of NALM-6 cells, cultures were selected in puromycin for seven days. Cells were then grown in the presence of the $IC_{30}$ concentration for each of the three PLK1 inhibitors. Most candidate genes were validated as enhancers of PLK1 inhibition and of these, *KIF18A* and *SKA1* were the most potent enhancers, in agreement with the primary screen results (S1 Fig).

## Other modifier genes of partial PLK1 inhibition

While centromere/kinetochore factors are clearly enriched among genetic enhancers of PLK1 inhibition, other genes with distinct functions were identified in our screens. Because PLK1 is also required for cytokinesis, we expected to recover genes involved in this process as enhancers of reduced PLK1 activity. *PRR14L* may be one such hit from our screen, where it ranked 7th among enhancers (S2 Table). Little is known about the molecular function of the PRR14L protein, but it was reported to localize at the cytokinetic midbody in HEK293T cells and interact with KIF4A and KIF23/MKLP1, two other midbody proteins that function in cytokinesis [31–34]. *KIF4A* ranked 87th among enhancers of PLK1 inhibition in our screen (S2 Table). Consistent with a role in cytokinesis, we found that silencing *PRR14L* in RPE-1 cells caused an increase in binucleated cells and enhanced binucleation upon treatment with BI2536 (S2 Fig). It remains to be discovered how PLK1 collaborates with PRR14L in cytokinesis.

Other enhancers we identified include stathmin (*STMN1*; ranked 3rd enhancer), a MT-depolymerization factor that was previously shown to promote mitotic entry upstream of PLK1 (Figs 1 and S1) [35]. The idea that partial inhibition of PLK1 sensitizes cells to the loss of positive regulators of mitotic entry is supported by the identification of *CCNB1* (21st enhancer), which encodes cyclin B1, a major mitotic cyclin that activates CDK1 [36]. We also found genes encoding several subunits of the dynactin complex including *ACTR1A* (4th

enhancer), *ACTR1B* (17th enhancer) and *DCTN2* (33rd enhancer). Dynactin is a regulator of the dynein motor protein which plays several roles in cell division [37], and genes encoding subunits of cytoplasmic dynein were also identified including *DYNLT1* (22nd enhancer) and *DYNC1LI1* (38th enhancer). Interestingly, dynactin promotes the targeting of PLK1 to kinetochores [38] and loss of this function may account for the enhancement of partial PLK1 inhibition. We also found *PDS5B* (5th enhancer), a positive and negative regulator of sister chromatid cohesion, which is proposed to be a direct target of PLK1 [39].

The two strongest suppressors of partial PLK1 inhibition identified are *XPR1* and *CCNF* (Fig 1 and S2 Table). *XPR1* encodes a transmembrane phosphate export protein [40]. *CCNF* encodes a component of the SCF$^{\text{Cyclin F}}$ E3 ubiquitin ligase with various roles in the cell cycle and was proposed to function as a tumor suppressor [41]. It will be interesting to explore if these genes are involved in negative regulation of PLK1 activity or PLK1-dependent functions.

## Inactivation of KIF18A or SKA1 in PLK1 inhibitor treated cells causes a mitotic arrest that depends on the spindle assembly checkpoint

We decided to investigate the mechanistic basis of the two strongest genetic enhancers of partial PLK1 inhibition that are *KIF18A* and *SKA1*. Both sgRNAs used for *KIF18A* targeted region coding for the N-terminal motor domain (S3A Fig) [42], while both sgRNAs for *SKA1* targeted the area coding for the C-terminal region proposed to interact with MTs and with the protein phosphatase PP1 (S3A Fig) [43]. To further validate *KIF18A* and *SKA1* as enhancers, we measured cell proliferation of each knockout NALM-6 population and control *AAVS1* and Azami-green populations in the presence of increasing concentrations of BI2536. Compared to controls, the *KIF18A* and *SKA1* knockout pools were more sensitive to the PLK1 inhibitor (Fig 2A and 2B). However, TIDE sequencing of the knockout populations revealed that disruption of the targeted *KIF18A* and *SKA1* regions was incomplete (S3B and S3C Fig) [44]. For KIF18A, we confirmed by Western blot that a protein of the expected wild type molecular weight remained at an intensity of approximately half that of control cells, suggesting that most *KIF18A*-targeted cells were heterozygous for the loss of function allele (S3D Fig). These results were in accordance with the previously established near-essentiality of the two genes in NALM-6 cells: *SKA1* is the 2693rd most essential gene, while *KIF18A* is the 2932nd most essential gene, such that retention of a wild type allele may be selected for in the knockout pools [27]. These results also indicate that inactivation of a single allele of either *KIF18A* or *SKA1* may be sufficient to sensitize cells to partial PLK1 inhibition.

KIF18A is a kinesin-8 motor protein with MT depolymerization activity that promotes chromosome alignment in metaphase by regulating MT dynamics near kinetochores [45–47]. SKA1 is a member of the SKA complex, which promotes stable kinetochore attachment to the spindle [43,48,49]. Despite their different molecular functions, inactivation of KIF18A or the SKA complex result in similar mitotic phenotypes, including chromosome congression and alignment defects in metaphase [46,47,50,51]. Because PLK1 promotes chromosome attachment and congression on the mitotic spindle, we hypothesized that PLK1 collaborates with KIF18A and SKA1 in mitosis. To test this possibility, we measured mitotic index by phospho-Histone H3 (pHH3, phosphorylated at Ser10) staining in *KIF18A* or *SKA1* knockout populations as a function of increasing concentrations of BI2536. As expected, high levels of PLK1 inhibition induced an increase in the mitotic index in all cultures. However, cells expressing sgRNAs against *KIF18A* or *SKA1* accumulated in mitosis at lower concentrations of BI2536 compared to control cells (Fig 3A and 3B). Similar results were obtained using GSK461364A as an alternative PLK1 inhibitor (S4A Fig). These results demonstrated that KIF18A and SKA1 are particularly crucial for mitotic progression when PLK1 function is compromised.

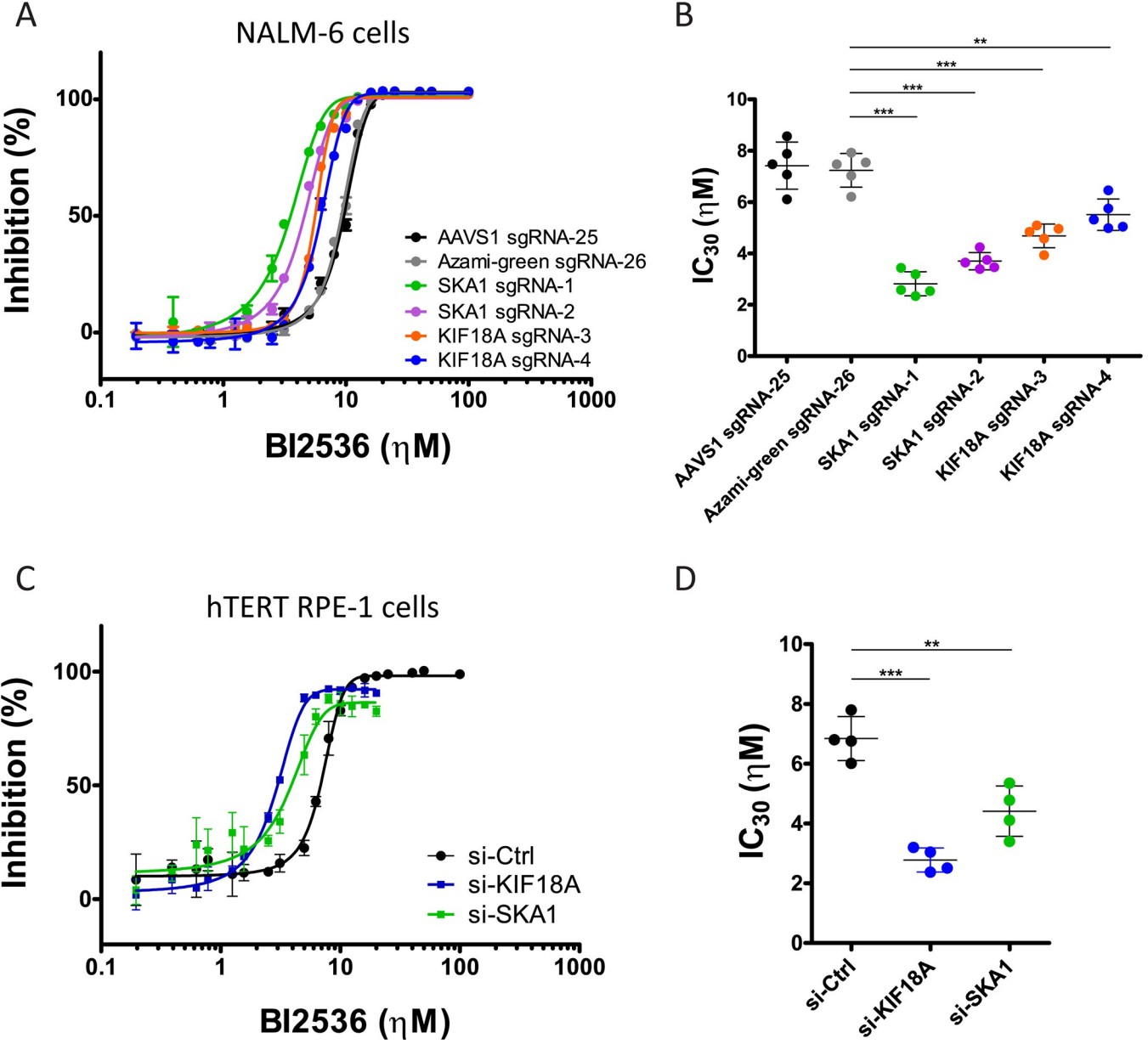

**Fig 2. Inactivation of KIF18A or SKA1 slows down proliferation of PLK1-compromised cells. A.** NALM-6 cells were selected for transduction with the indicated sgRNA expression constructs and cultures were analyzed for proliferation using Alamar Blue after 96 hours in the presence of different concentrations of BI2536. Results from one representative experiment are shown. Data points show averages of triplicates. Error bars: SD. **B.** BI2536 IC$_{30}$ values for inhibition of proliferation obtained in experiments as in A. Averages of five independent experiments are shown ±SD. **C.** hTERT RPE-1 cells were transfected twice with the indicated siRNAs and cultures were analyzed for proliferation using Alamar Blue after 96 hours in the presence of different concentrations of BI2536. Results from one representative experiment are shown. Data points show averages of triplicates. Error bars: SD. **D.** BI2536 IC$_{30}$ values for inhibition of proliferation obtained in experiments as in C. Averages of four independent experiments are shown ±SD. *** p < 0.001, ** p < 0.01 in Student's unpaired T test. Coordinate values used to generate graphs are available in S2 Data.

NALM-6 pre-B lymphocytic leukemia cells are small and non-adherent, which made detailed characterization of mitotic phenotypes challenging. We sought to validate our PLK1 genetic dependencies in a different cell line more suitable for the examination of mitotic phenotypes. To this end, we used Retinal Pigmental Epithelial-1 cells immortalized by the expression of hTERT (hTERT RPE-1), which are commonly used to study mitosis [52]. We initially

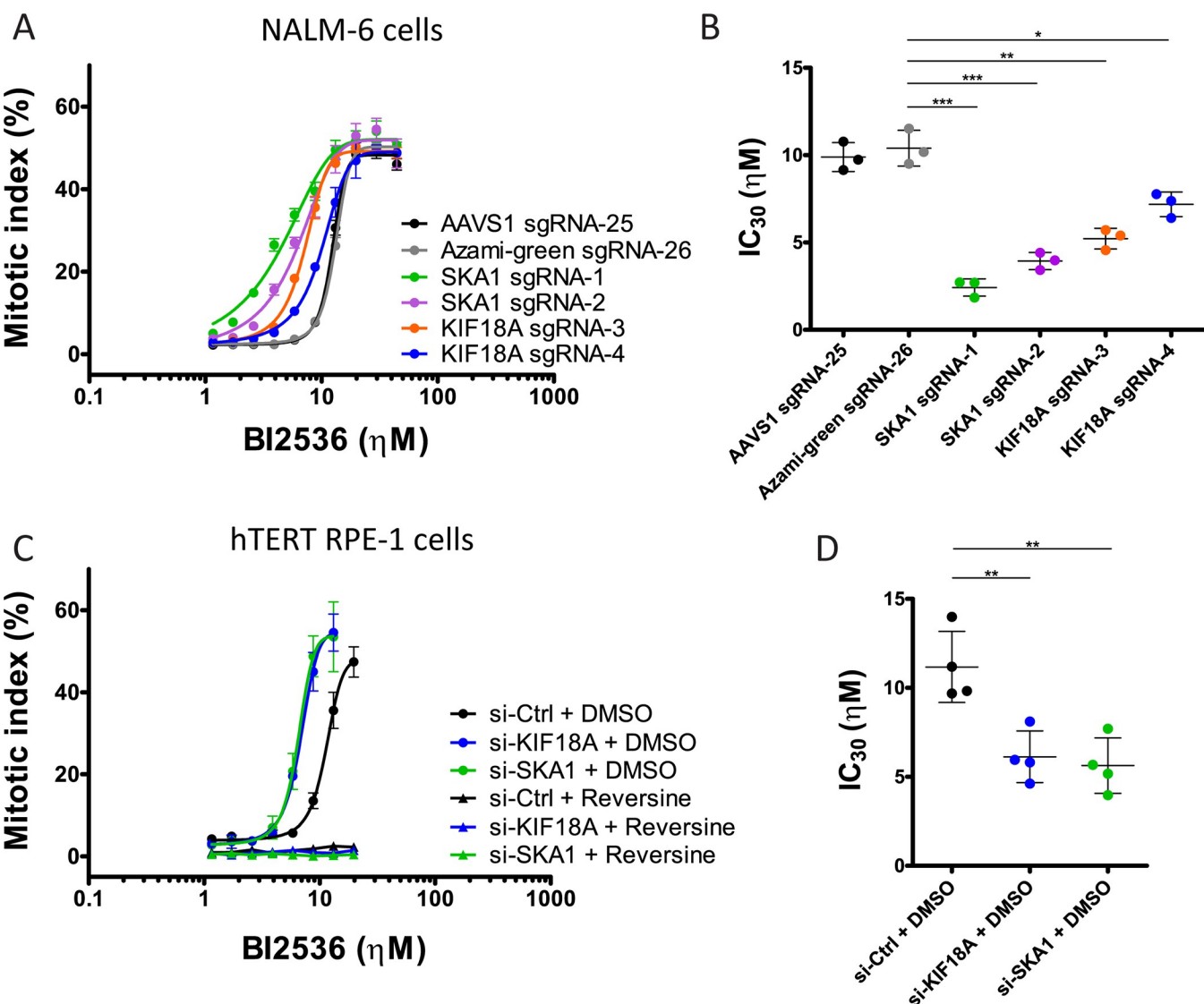

**Fig 3. Inactivation of KIF18A or SKA1 sensitizes PLK1-compromised cells to a SAC-dependent mitotic arrest. A.** Targeting *KIF18A* or *SKA1* by CRISPR sensitizes NALM-6 cells to the BI2536-induced mitotic arrest. The mitotic index was measured after immunofluorescence for pHH3. A representative experiment is shown. Data points are averages of triplicates ±SD. **B.** $IC_{30}$ values for BI2536 from 3 independent experiments as in A. Averages ±SD are shown. **C.** Silencing KIF18A or SKA1 by siRNA sensitizes hTERT RPE-1 cells to the BI2536-induced, SAC-dependent mitotic arrest. Inhibition of MPS1 with reversine strongly abrogates the mitotic arrest. The mitotic index was measured after immunofluorescence for pHH3. Data points are averages of triplicates ±SD from a representative experiment. **D.** $IC_{30}$ values for BI2536 from 4 independent experiments as in C. Averages ±SD are shown. *** $p < 0.001$, ** $p < 0.01$, * $p < 0.05$ in Student's unpaired T test. Coordinate values used to generate graphs are available in S3 Data.

generated RPE-1 culture populations in which *KIF18A* or *SKA1* was targeted by the same sgRNAs used above and tested for responses to increasing concentrations of BI2536. However, we obtained variable and inconclusive results, possibly because *KIF18A* and *SKA1* are the 411[th] and 662[nd] most essential genes in hTERT RPE-1 cells [53]. As an alternative approach, we used siRNAs to silence *KIF18A* and *SKA1* in hTERT RPE-1 cell populations. Transfection of siRNAs against either *KIF18A* or *SKA1* rendered hTERT RPE-1 cells significantly more sensitive to BI2536 than control cells in both the proliferation assay (Figs 2C, 2D and S5) and the mitotic index assay (Fig 3C and 3D). From these results, we conclude that the sensitivity of

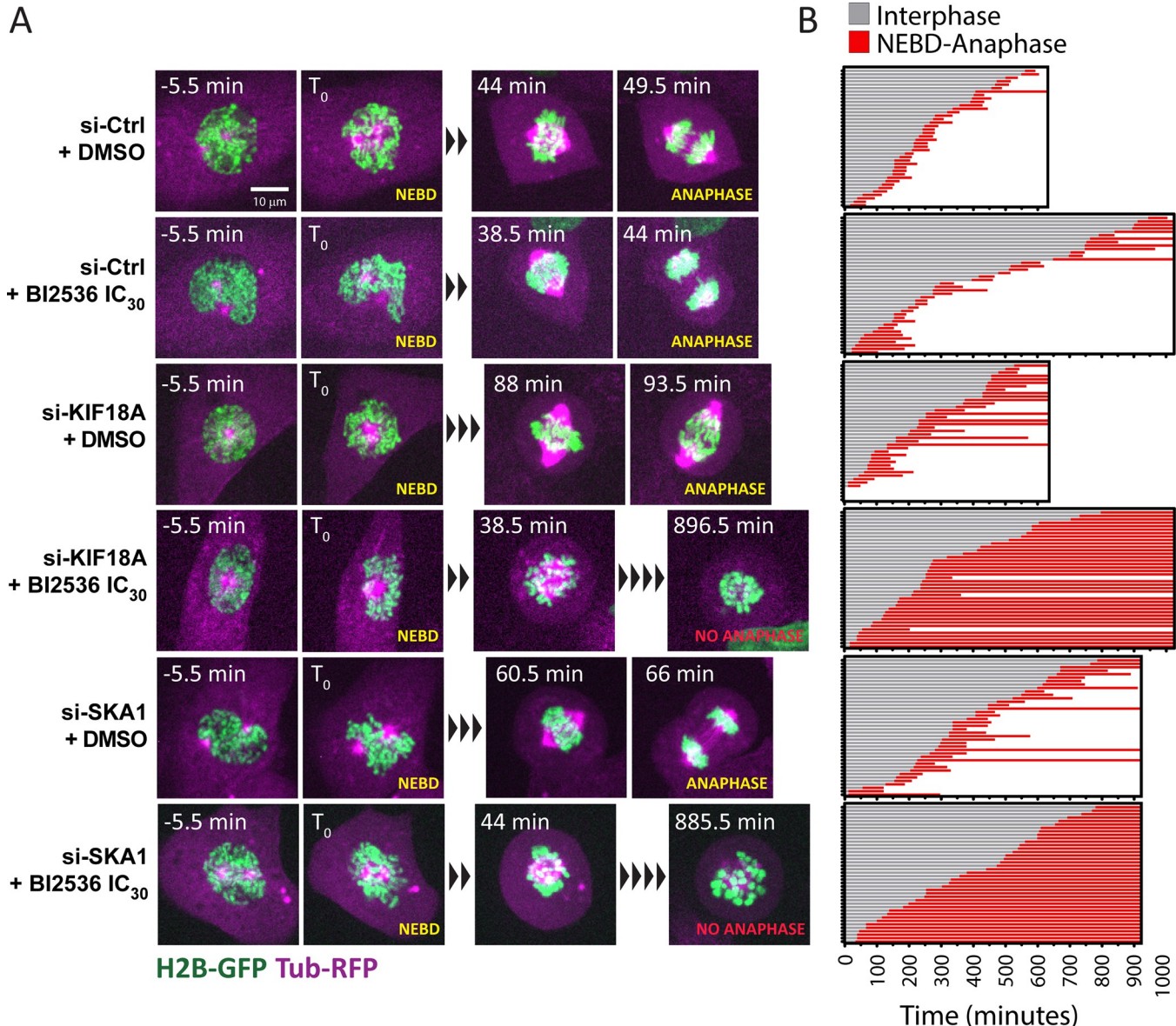

**Fig 4. Inactivation of KIF18A or SKA1 sensitizes PLK1-compromised cells to a metaphase arrest.** A. Time-lapse images of RPE-1 cells expressing H2B-GFP and α-Tub-mRFP and treated as indicated. BI2536 was added at the $IC_{30}$ concentration (5 nM). The time of Nuclear Envelope BreakDown (NEBD) was set as time zero. Representative cells are shown at selected time points of interest. B. Mitotic history profiles of individual cells treated and imaged as indicated in A. Grey lines indicate the time before mitotic entry. Red lines indicate the time between NEBD and anaphase. Between 36 and 40 cells were analyzed for each condition. Coordinate values used to generate graphs are available in S4 Data.

PLK1-inhibited cells to inactivation of *KIF18A* or *SKA1* reflects fundamental functional interdependencies of PLK1 and KIF18A or SKA1 in mitosis.

We next imaged mitotic progression using hTERT RPE-1 cells expressing Histone 2B-GFP (H2B-GFP) and α-Tubulin-mRFP (α-Tub-mRFP) (Fig 4A and 4B and S1–S6 Videos). The time in mitosis between Nuclear Envelope Breakdown (NEBD) and anaphase was monitored. NEBD was determined as the point when α-Tub-RFP enters the nuclear area, and anaphase onset was visualized by separation of H2B-GFP marked chromosomes. Treatment with BI2536 at the $IC_{30}$ concentration alone had little effect on mitotic duration, which typically

lasted less than one hour. Depletion of KIF18A alone delayed completion of mitosis but most cells ultimately completed division. By contrast, combining both treatments caused most cells to arrest in prometaphase/metaphase for several hours, and most cells that entered mitosis did not complete mitosis even after 16 h. Similar results were obtained with SKA1 as partial PLK1 inhibition combined with SKA1 depletion caused a near completely penetrant mitotic arrest.

We hypothesized that the mitotic arrest observed in PLK1-inhibited cells depleted for KIF18A or SKA1 may be due to activation of the Spindle Assembly Checkpoint (SAC). To test this model, we inhibited the MPS1 kinase, which is required for the SAC, using the competitive MPS1 inhibitor reversine [54]. We found that SAC inhibition strongly abrogated the mitotic arrest due to BI2536 treatments, even when KIF18A or SKA1 were depleted (Fig 3C). Consistent with this observation, the primary PLK1 screens identified several genes that encode proteins required for the SAC as suppressors of partial PLK1 inhibition. These genes included *BUB1B* (encodes BUBR1, ranked 3rd suppressor), *MAD1L1* (encodes MAD1, ranked 11th) and *MAD2L1* (encodes MAD2, ranked 191st) (Fig 1C and S2 Table). The screens also identified the *TTK* gene (ranked 22nd), which encodes MPS1, the direct target of reversine. These results indicate that the mitotic arrest and reduced proliferation observed when PLK1 and KIF18A or SKA1 functions are compromised are at least in part due to activation of the SAC.

## Inactivation of KIF18A or SKA1 enhances nuclear reassembly defects in PLK1-compromised cells

The SAC-dependent mitotic arrest obtained when KIF18A or SKA1 is depleted and PLK1 is partially inhibited suggested that cells may fail to attach and align chromosomes on the mitotic spindle. To test this hypothesis, we used immunofluorescence with Anti-Centromere Antibodies (ACA) and measured the dispersion of centromeres along the spindle (see Materials and Methods). We found that silencing of *KIF18A* increased the dispersion of centromeres in metaphase, as previously shown [47]. However, partial inhibition of PLK1 had no effect and the combination of both perturbations did not enhance centromeric dispersion relative to KIF18A depletion alone (Fig 5A and 5B). We observed that bipolar spindles were longer upon KIF18A depletion, also consistent with previous findings [46,47]. Conversely, spindles became slightly shorter upon partial inhibition of PLK1, again consistent with previous observations [55]. Combining PLK1 inhibition with KIF18A depletion resulted in spindle lengths similar to those observed after depletion of KIF18A alone (Fig 5A and 5C). Similar impacts on centromere dispersion and spindle length were obtained with the depletion of SKA1, alone or in combination with partial inhibition of PLK1 (Fig 5A–5C). SKA1 depletion increased the occurrence of multipolar spindles, which could contribute to increase centromere dispersion as quantified along the longest axis of the spindle. We concluded that the combined inactivation of KIF18A and PLK1, or SKA1 and PLK1 did not cause additive defects in the configuration of the pre-anaphase chromosome/spindle apparatus.

Although combined KIF18A and PLK1 inactivation caused most cells to arrest in mitosis before anaphase, we found that a fraction of cells entered anaphase after a prolonged arrest. These cells tended to acquire micronuclei after mitosis, while cells treated with BI2536 alone could divide without micronucleation (Fig 6A and S7 and S8 Videos). Staining for DNA (DAPI) and Lamin A revealed that treatment with BI2536 at the IC$_{30}$ value together with siRNA-mediated depletion of KIF18A caused structural nuclear defects suggesting problems during nuclear reassembly (the formation of a single nucleus with a normal shape after mitosis). These defects included multilobed nuclei and micronuclei completely separated from the main nuclear mass (Fig 6B). To quantify this phenotype, we measured the solidity and

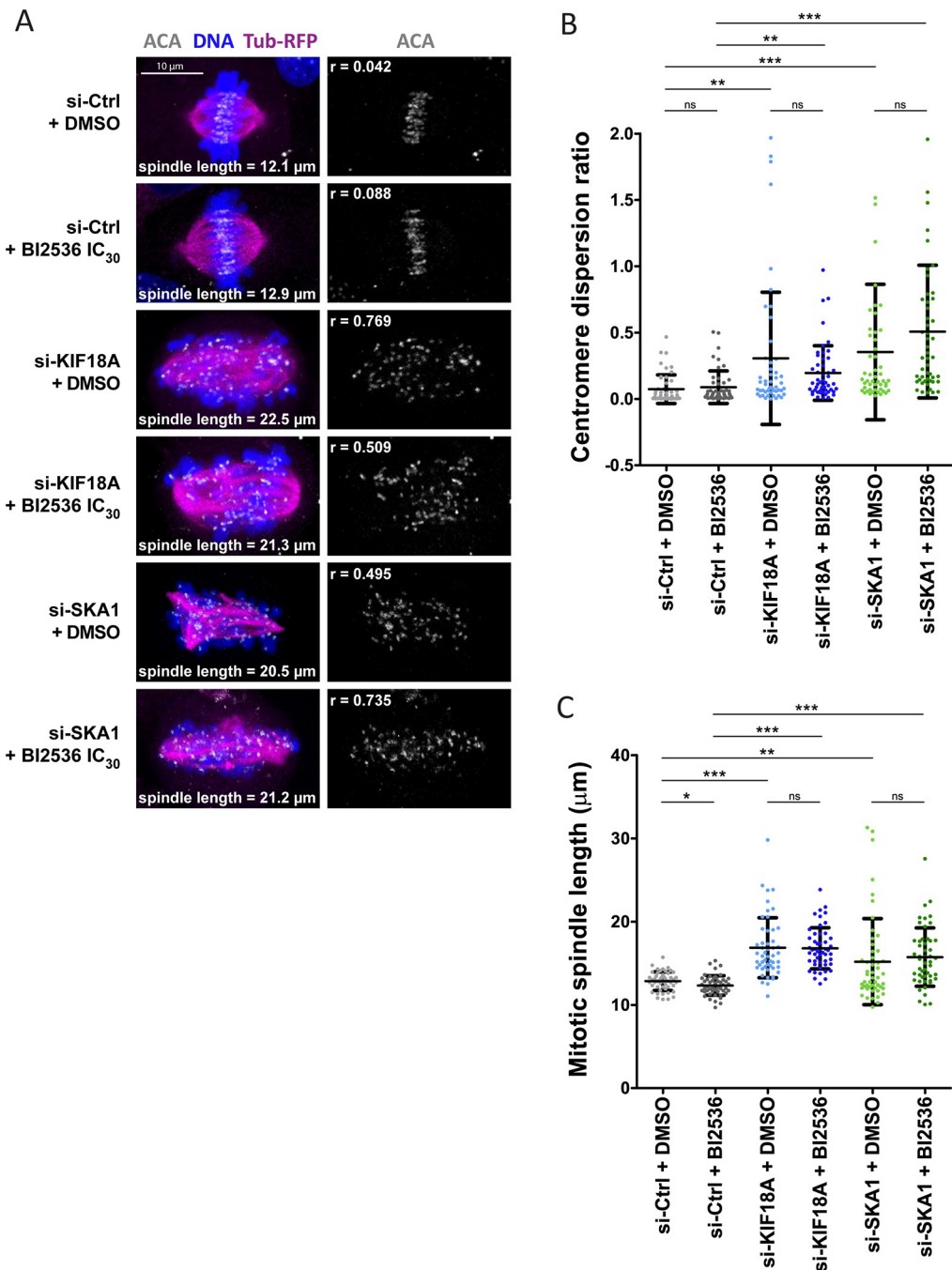

**Fig 5. Inactivation of KIF18A or SKA1 causes spatial mitotic defects before anaphase that are not grossly enhanced by partial inhibition of PLK1. A.** RPE-1 cells expressing H2B-GFP and α-Tub-mRFP (magenta) were treated as indicated, fixed and stained for centromeres (ACA, white) and DNA (DAPI, blue). BI2536 was added at the IC$_{30}$ concentration (5 nM). Representative cells are shown, with their measured spindle lengths and centromere dispersion ratios (r). **B-C.** Quantification of centromere dispersion ratios (B) and mitotic spindle lengths (C) of cells treated as indicated and analyzed as in A. Fifty cells were analyzed per condition. Averages ±SD are shown. *** p < 0.001, ** p < 0.01, * p < 0.05 in Student's unpaired T test. ns: non-significant. Coordinate values used to generate graphs are available in S5 Data.

circularity of the nuclear shapes as defined by the DNA mass (see Materials and Methods) (Fig 6C and 6D). We found that nuclear defects were enhanced in cells in which PLK1 and KIF18A were compromised, as compared to the single treatments alone. These results suggested that simultaneously compromising PLK1 and KIF18A functions also enhanced the accumulation of post-mitotic structural nuclear defects. We observed that a similar enhancement of nuclear defects was caused by the combined inactivation of PLK1 and SKA1 (Fig 6B–6D). The similarity of these phenotypes, despite the different molecular functions of KIF18A and SKA1, suggested that partial loss of PLK1 function sensitized cells to different types of kinetochore perturbations.

## CENP-A recruitment is highly sensitive to partial inhibition of PLK1

We sought to identify the precise function of PLK1 that renders cells particularly sensitive to different kinetochore perturbations upon partial inhibition of PLK1. It is established that PLK1 is required for CENP-A recruitment during the mitosis/G1 transition [56]. PLK1 promotes the centromeric localization of the MIS18 complex which in turn recruits HJURP, a CENP-A chaperone required for its centromeric loading [56–61]. As described above, genes encoding centromeric proteins were predominant genetic enhancers of partial PLK1 inhibition. In particular, our PLK1 screens identified CENP-A (ranked 42nd), MIS18B/OIP5 (ranked 59th) and HJURP (ranked 80th) as enhancers. Therefore, we wondered if CENP-A loading was particularly sensitive to PLK1 inhibition. To test this model, we treated RPE-1 cells with various concentrations of BI2536 for 72 h and quantified CENP-A levels on DNA by immunofluorescence (Fig 7A and 7B). Strikingly, we found that concentrations of PLK1 inhibitor as low as the $IC_{10}$ (in cell proliferation) lead to a marked decrease in CENP-A levels on DNA. Consistent results were obtained by Western blots showing CENP-A levels on chromatin following cell fractionation (Fig 7C). These results suggest that CENP-A loading is extremely sensitive to partial PLK1 inhibition. Compromised centromere assembly may therefore underlie the sensitivity of PLK1-compromised cells to different KT perturbations.

## Discussion

### Kinetochore functions depend on high PLK1 activity

The many genes encoding centromere/kinetochore factors identified as enhancers in our screens strongly suggest that the most critical functions of PLK1 impact chromosome attachment and segregation on the mitotic spindle. These genetic results are consistent with a previous study that used chemical genetics to titrate PLK1 activity in RPE-1 cells [55]. The precise substrates of PLK1 at centromeres and kinetochores are numerous, and how their combined regulation functions to coordinate chromosome dynamics in mitosis is a complex problem [13,62]. *KIF18A* was the strongest enhancer of PLK1 identified in our genetic screens. The kinesin motor protein encoded by *KIF18A* is concentrated near KTs to regulate the oscillations of chromosomes attached on the spindle, thereby promoting chromosome alignment before anaphase [46,47]. Loss of function of KIF18A was previously shown to result in less compact chromosome masses during anaphase, leading to nuclear reassembly defects including micronuclei [63,64]. On the other hand, partial inhibition of PLK1 results in lagging chromosomes in anaphase leading to micronucleation [55]. Our finding that combined inactivation of PLK1 and KIF18A increases nuclear reassembly defects compared to individual treatments likely reflects the requirement of these proteins for synchronous chromosome segregation. Similarly, our observation that loss of SKA1 in PLK1-compromised cells enhances nuclear reassembly defects is consistent with the known function of the SKA complex in promoting stable kinetochore attachments [43,48–50,65]. Since we did not observe enhancement of centromere

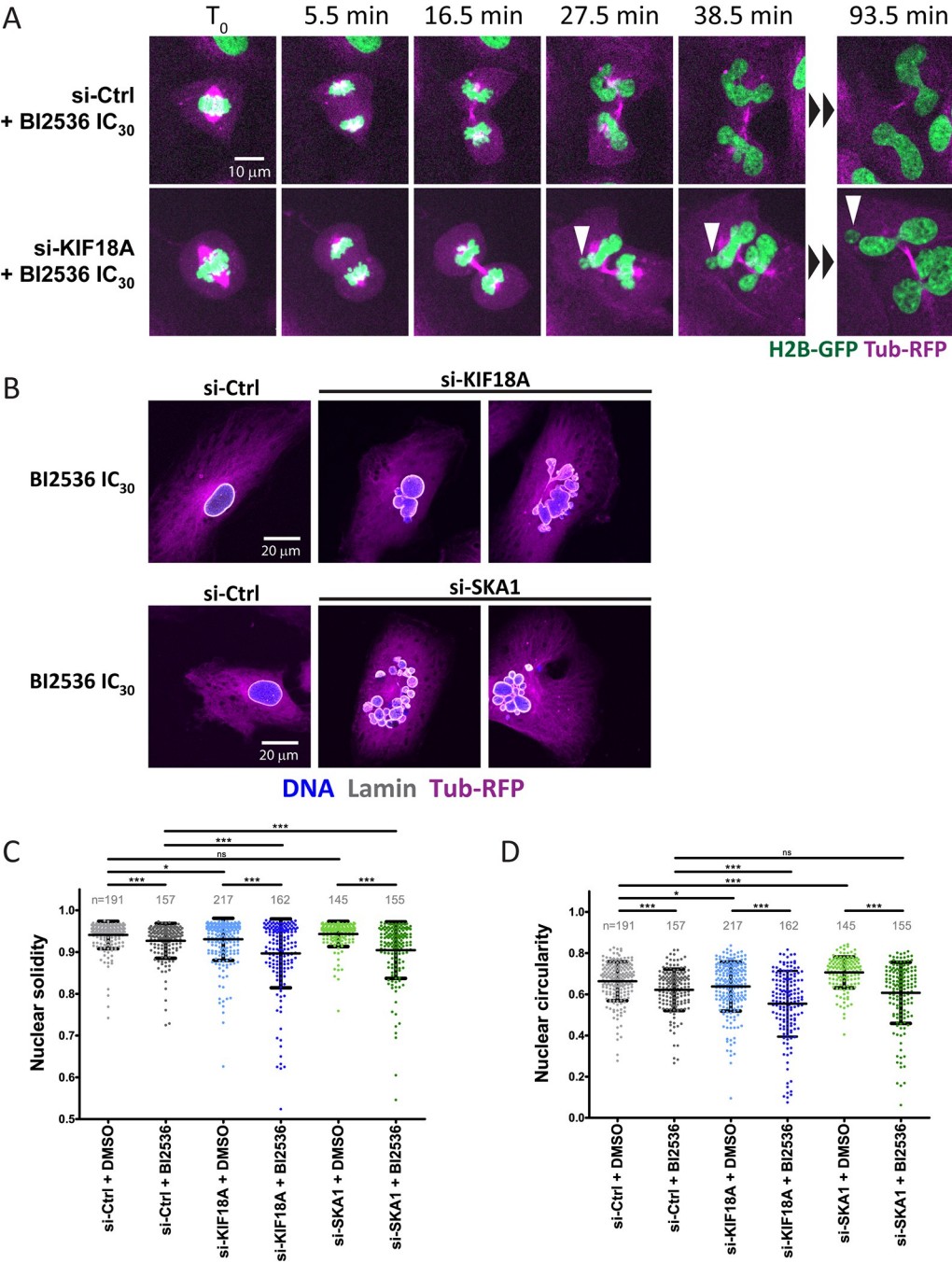

**Fig 6. Inactivation of KIF18A enhances nuclear reassembly defects after mitosis in PLK1-compromised cells. A.** Time-lapse images of RPE-1 cells expressing H2B-GFP and α-Tub-mRFP and treated as indicated. BI2536 was added at the $IC_{30}$ concentration (5 nM). The top cell treated with BI2536 and a non-target siRNA reassembled the nucleus normally. The bottom cell treated with BI2536 and siRNA against KIF18A assembled micronuclei (arrowheads) after mitosis. The time prior to anaphase onset was set as time zero. **B.** RPE-1 cells expressing H2B-GFP and α-Tub-mRFP and treated as indicated were fixed and stained for Lamin A (white) and DAPI (blue). Note the structural nuclear defects occurring when KIF18A or SKA1 is depleted in the presence of BI2536 at the $IC_{30}$. **C-D.** Quantification of solidity (C) and circularity (D) of the nucleus in cells treated as indicated. Averages ±SD are shown. *** $p < 0.001$, ** $p < 0.01$, * $p < 0.05$ in Student's unpaired T test. ns: non-significant. Coordinate values used to generate graphs are available in S6 Data.

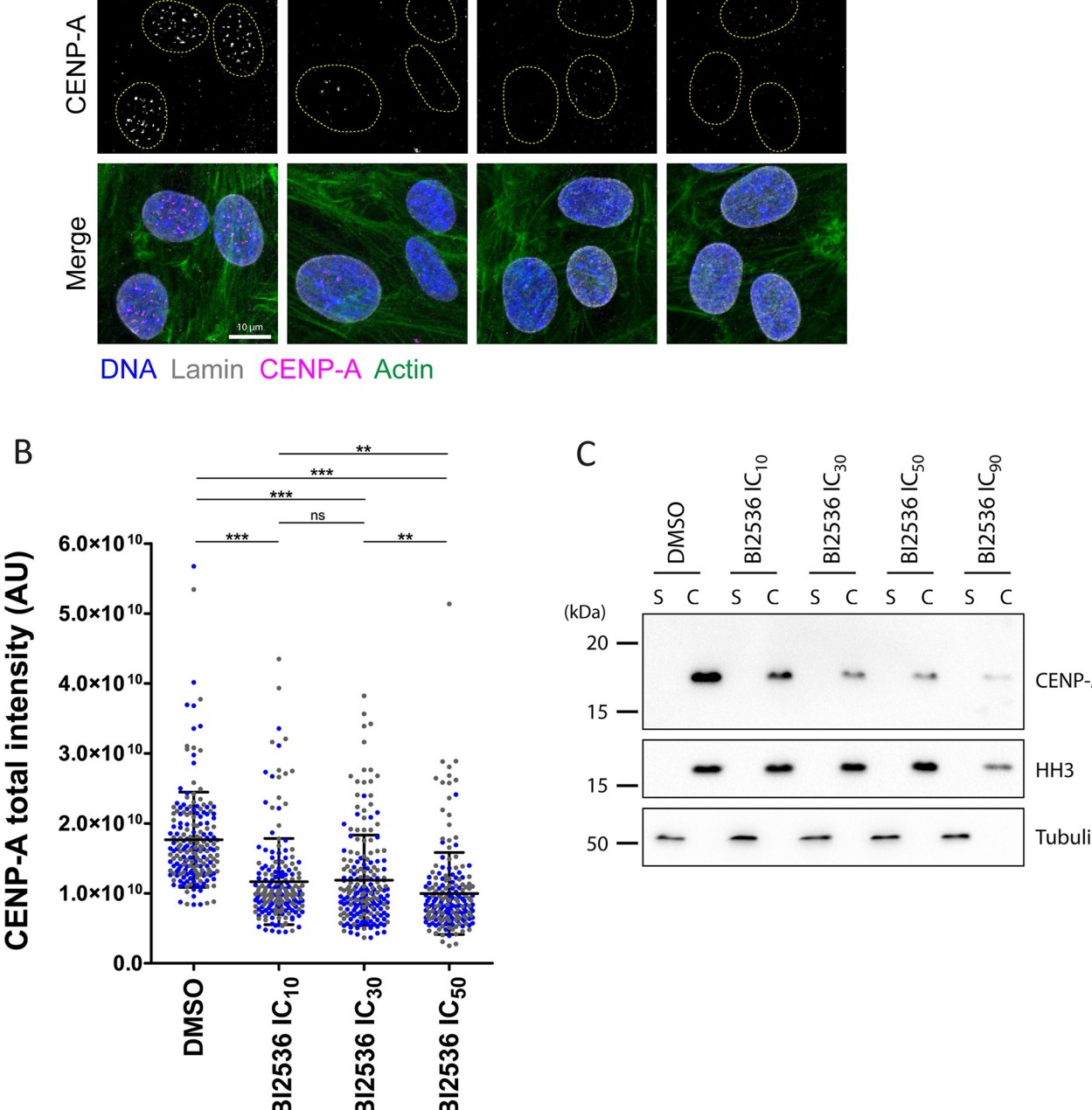

**Fig 7. CENP-A loading on centromeres is extremely sensitive to partial PLK1 inhibition. A.** Immunofluorescence in RPE-1 cells treated for 72 h with the indicated concentrations of BI2536. CENP-A levels were measured in the DAPI-stained areas (inside yellow dotted lines). **B.** Quantifications as illustrated in A. Each data point is from a single cell. Blue and grey points are from two independent experiments (100 cells were analyzed per condition per experiment). Averages ±SD are shown. *** p < 0.001, ** p < 0.01 in Student's unpaired T test. ns: non-significant. **C.** Western blots showing CENP-A levels in a crude chromatin fraction (C, pellet) vs the soluble fraction (S). The crude chromatin fraction was resuspended in 1/10 volume relative to the total lysate. Histone H3 (HH3) and α-Tubulin were monitored as chromatin and soluble markers, respectively. Concentrations of BI2536 used were measured in a cell proliferation inhibition assay (see Materials & Methods; $IC_{10}$: 2.7 nM, $IC_{30}$: 5.0 nM, $IC_{50}$: 7.2 nM. $IC_{90}$: 19.2 nM). Coordinate values used to generate graphs are available in S7 Data.

dispersion when PLK1 was inhibited in addition to KIF18A or SKA1 depletion, defects in congression before anaphase are unlikely to be the underlying cause of the enhanced mitotic arrests and nuclear reassembly defects. Instead, weaker kinetochore attachments may enhance defects specifically in anaphase. Nuclear reassembly defects may also arise as cells exit the prolonged mitotic arrest by slippage, potentially leading to the reassembly of defective nuclei from unsegregated chromosomes.

We found that CENP-A loading is extremely sensitive to partial PLK1 inhibition. Indeed, most CENP-A was depleted from DNA at concentrations of PLK1 inhibitor that only slightly hampered cell proliferation. This suggests that in the presence of low concentrations of PLK1 inhibitor cells can proliferate with reduced levels of CENP-A at centromeres. Because CENP-A is required for the recruitment of other centromeric proteins and kinetochore factors [66,67], reduced CENP-A levels at centromeres may result in compromised KT structure and function, making cells hypersensitive to further KT perturbations, particularly depletion of KIF18A or SKA1.

The predominance of centromere/KT factors among enhancers of partial PLK1 inhibition may also be explained by a reduction in the ability of centromere/KT structures to recruit PLK1. Recently, it has been reported that PLK1 is recruited to kinetochores by two parallel receptors: one receptor is the complex formed by CENP-U, CENP-Q, CENP-O and CENP-P, and the other receptor is the mitotic checkpoint kinase BUB1 [68–70]. Strikingly, we also identified all of these proteins as enhancers of partial PLK1 inhibition (ranked 10th, 18th, 26th, 81th and 41th enhancers, respectively). These genetic dependencies suggest that the localization of PLK1 at KTs may be critical for PLK1 function in mitosis. We suggest that multiple mechanisms in the assembly and/or function of centromeres and kinetochores are likely to be compromised when PLK1 is partially inhibited, thereby rendering cells hypersensitive to additional kinetochore perturbations, as epitomized by the loss of KIF18A or SKA1.

### Implications for cancer therapies targeting PLK1

KIF18A has recently emerged as a promising cancer drug target. Cancer cells that are aneuploid, tetraploid or have a chromosomal instability phenotype are extremely vulnerable to the loss of KIF18A, as compared to euploid cancer cells [71–73]. This sensitivity closely parallels the vulnerability of such cancer cells to SAC inhibition [74]. The sensitivity of cancer cells with severe karyotypic abnormalities to KIF18A or SAC inactivation may be due to increased burden of supernumerary chromosomes on assortment and segregation in mitosis. Our finding that KIF18A, the SAC and PLK1 are highly interdependent suggests that PLK1 may also be a target of choice in cancers with complex karyotypes. Indeed, it has been reported that tetraploid cancer cells are hypersensitive to PLK1 inhibition [75,76]. We note that small molecule inhibitors of KIF18A are being developed as a novel anti-cancer strategy [77,78]. We propose that combining PLK1 inhibitors with KIF18A inhibitors may prove particularly effective in treating cancers with complex karyotypes.

The MT stabilizer taxol (paclitaxel) is one of the most successful and widely used anti-cancer drugs [79,80]. Although its anti-tumor actions are not completely understood, evidence suggests that taxol may lead to tumor regression predominantly by inducing pro-inflammatory micronucleation resulting from aberrant mitotic exit [81,82]. Interestingly, we found that the combined inactivation of PLK1 and KIF18A (or SKA1) leads to enhanced nuclear reassembly defects including widespread micronucleation. We suggest that combined inhibition of PLK1 and KIF18A may mimic the relevant action of taxol without collateral toxicity due to MT poisoning in interphase cells.

## Materials & methods

### Cell culture, siRNA and inhibitors

NALM-6, HEK293T and hTERT RPE-1 cells were cultured at 37°C with 5% $CO_2$; in RPMI medium (Wisent) for NALM-6 and in DMEM medium (Wisent) for HEK293T and RPE-1 cells, all supplemented with 10% fetal bovine serum (FBS, HyClone), penicillin and streptomycin (Wisent). SMARTpool ON-TARGETplus human KIF18A siRNA, human SKA1 siRNA, human PRR14L siRNA and ON-TARGETplus Non-targeting Pool were purchased from Horizon Discovery. SiRNA transfections in RPE-1 cells were carried out with Lipofectamine RNAi-MAX Transfection Reagent (Invitrogen) according to the manufacturer's procedures. BI2536, BI6727/Volasertib and GSK461364A were purchased from MedChemExpress and Reversine was purchased from Cayman Chemical Co.

### Plasmids and CRISPR cell lines

Two of the ten sgRNA per gene from the EKO library used in the screen were selected with two sgRNA controls (*AAVS1* and Azami-green) for the knock-out validation in NALM-6 cells. Complementary oligonucleotides corresponding to coding sequences were purchased from IDT (Integrated DNA Technologies). Oligonucleotides were phosphorylated, annealed and ligated into LentiCRISPRv2 (Addgene #52961) digested by BsmB1 (New England Biolabs). Plasmids were transfected into HEK293T cells using polyethylenimine along with the lentiviral packaging plasmid psPAX2 (Addgene #12260) and the envelope plasmid pCMV-VSV-G (Addgene #8454). Sixteen hours post-transfection, media were changed for DMEM with 2% FBS to boost viral particle production. Forty-eight hours post-transfection, viral particles were harvested and filtered through a 0.45 μm membrane filter. For lentiviral transduction, NALM-6 cells were incubated with viral particles and protamine sulfate (final concentration 10 μg/mL; Sigma Aldrich) for 48 h. Cells were then subjected to puromycin (final concentration 1 μg/mL) selection for 7 days.

### Proliferation inhibition assays

To determine the $IC_{10}$, $IC_{30}$ and $IC_{50}$ values of the 3 PLK1 inhibitors for use in the pooled CRISPR knockout screens, a NALM-6 clone bearing an integrated inducible Cas9 expression cassette generated by lentiviral infection with plasmid pCW-Cas9 (Addgene #50661) was seeded in 384-well plates at 200,000 cell per mL and 50 μL per well. Twenty 2.5 nL drops of PLK1 inhibitor diluted in DMSO or DMSO solvent only were added in triplicate with an Echo 555 acoustic liquid handler (Labcyte) for a total of 50 nL in each well to generate a gradient of PLK1 inhibitor concentration. After 3 days growth at 37°C in a high humidity chamber, 25 μL of media was removed from the top of cells and 25 μL of CellTiter-Glo Luminescent cell viability substrate (Promega) was added to each well and luminescence read on a BioTek Synergy/ Neo microplate reader (Promega). Statistical analysis was performed with the IDBS Activity Base software (IDBS).

For knock-out validation in NALM-6, the *AAVS1* sgRNA-25 control cell line was analyzed for proliferation after 96 h of treatment with increasing concentrations of the 3 PLK1 inhibitors. Cells were counted using a Beckman-Coulter Z2 cell counter and data plotted in Graph-Pad to determine the $IC_{30}$ concentration (indicated in S1 Fig). hTERT RPE-1 cells, either wild type or expressing H2B-GFP and α-Tub-mRFP, were also analyzed in this manner to determine the $IC_{10}$ (2.7 nM), $IC_{30}$ (5.0 nM), $IC_{50}$ (7.2 nM) and $IC_{90}$ (19.2 nM) concentrations of BI2536 used in experiments shown in Figs 4, 5, 6, 7, S2 and S5).

## Genome-wide CRISPR knockout screens

The NALM-6 pCW-Cas9 clone described above was transduced with the EKO sgRNA library to perform the genome-wide knockout screens [27]. A frozen aliquot of the uninduced library pool was thawed in RPMI 1640 media containing 10% FBS (v/v) and Cas9 expression was induced with 2 μg/mL doxycycline. After 7 days of doxycycline treatment, the pooled library was split in different T-75 flasks ($28 \times 10^6$ cells per flask, corresponding to 100 cells/sgRNA for the 278,754 different sgRNAs in the EKO library) in 70 mL at $4 \times 10^5$ cells/mL. Compounds BI2536, BI6727 or GSK461364A were added at 1000X to a final DMSO concentration of 0.1% (v/v). Cells were counted with a Beckman Coulter Z2 cell counter every 2 days, and upon reaching $8 \times 10^5$ cells/mL, cells were reseeded at $4 \times 10^5$ cells/mL in the presence of fresh compound for a total of 8 days. After compound treatment, cells were collected, genomic DNA was extracted using a Gentra Puregene Cell kit according to manufacturer's instructions (QIAGEN), and sgRNA sequences PCR-amplified as described [27].

SgRNA frequencies were obtained by next-generation sequencing (Illumina HiSeq2000 or NextSeq 500, as indicated in NCBI GEO record GSE228155). Reads were aligned using Bowtie2.2.5 [83] in the forward direction only (–norc option) with otherwise default parameters and total read counts per sgRNA tabulated. Read counts from the different sequencing lanes for the same sample were summed and read counts from all control samples and time-points from each experiment were summed (for further details see NCBI GEO record GSE228155). Control samples were pooled irrespective of whether they were treated with DMSO 0.1% (v/v) or untreated (i.e., no DMSO), as no DMSO-specific signature was observed and pooling together read counts from all controls provided additional statistical power. Context-dependent chemogenomic interaction scores were calculated using a modified version of the RANKS algorithm [27], which uses guides targeting similarly essential genes as controls to distinguish condition-specific chemogenomic interactions from non-specific fitness/essentiality phenotypes.

## Indel sequencing

For assessment of indel frequency, genomic DNA was extracted using DNAzol (Invitrogen), followed by PCR amplification of the loci of interest. Products were purified and analyzed by Sanger sequencing. The Synthego ICE Analysis tool online (2019. v3.0. Synthego) was used to analyze the indel decomposition of the sequencing traces using the TIDE method [44].

## RT-qPCR

Cells were collected and pelleted after the indicated treatments. RNA was extracted using a RNeasy mini kit (QIAGEN), total RNA was treated with DNase, and then reverse transcribed using the Maxima First Strand cDNA synthesis kit with dsDNase (Thermo Fisher Scientific). Gene expression was determined using assays designed with the Universal Probe Library (UPL probe) from Roche (www.universalprobelibrary.com). For each qPCR assay, a standard curve was generated to ensure that the efficiency of the assay was between 90% and 110%. Oligonucleotides used in this study and related information is presented in S3 Table. The QuantStudio qPCR instrument (Thermo Fisher Scientific) was used to detect the amplification level. Relative expression comparison ($RQ = 2^{-\Delta\Delta CT}$) was calculated using the Expression Suite software (Thermo Fisher Scientific), using the housekeeping genes HPRT and ACTB as controls for the normalization.

## Viability assays

For NALM-6 knock-out cell lines, 10,000 cells/well were seeded in 96-well, black, flat-bottom plates in triplicate. Cells were then treated for 96 h with BI2536 at the indicated concentrations. For RPE-1, cells were trypsinized 48 h post-transfection and transfected with the siRNA a second time. Immediately afterwards, 1000 cells/well were seeded in 96-well black plate in triplicate and treated for 96 h with the BI2536 at the indicated concentrations. Twenty μL of a 0.1% AlamarBlue (Resazurin sodium salt; Sigma-Aldrich) solution was added to each well to obtain a final concentration of 0.01%. Plates were incubated 4 h at 37˚C and then fluorescence was monitored from the top of the wells using a Tecan plate reader using 530 nm excitation and 580 nm emission wavelengths. To normalize the cell viability before compound treatments, NALM-6 knockout cell lines and siRNA-treated RPE-1 cells were seeded as described above, AlamarBlue was added immediately for 4 h and fluorescence was read. This basal value obtained for each NALM-6 knockout cell line and RPE-1 siRNA treatment was then subtracted from the fluorescence values obtained after 96 h of treatment with BI2536. Fluorescence values for medium alone was also subtracted.

## Immunofluorescence and Western blotting

Immunofluorescence and Western blotting were performed essentially as previously described [84]. For immunofluorescence, the following antibodies were used. Primary antibodies: anti-Lamin A (1:1000; Sigma-Aldrich), anti-centromere protein (ACA, CREST, 1:400; Antibodies Incorporated), anti-phospho-Histone H3 (Ser10, 1:300; Sigma-Aldrich), anti-α-tubulin DM1A (1:500; Sigma-Aldrich), anti-CENP-A (3–19, 1:300; Invitrogen). Secondary antibodies: Alexa Fluor 647 Goat Anti-Rabbit IgG (H+L, 1:200; Invitrogen), Alexa Fluor 647 IgG (H+L) Cross-Adsorbed Goat anti-Human (1:200; Invitrogen), Alexa Fluor 488 goat anti-rabbit (1:200; Invitrogen), Texas Red-X goat anti-mouse IgG (1:200; Invitrogen) and Alexa Fluor 488 Phalloidin (to label F-actin, 1:1000; Invitrogen). For Western blotting, the following primary antibodies were used: anti-α-tubulin DM1A (1:5000; Sigma-Aldrich), anti-KIF18A (1:1000; Bethyl Laboratories Inc.), anti-CENP-A (3–19, 1:2000; Invitrogen) and anti-Histone H3 (3H1, 1:1000; New England BioLabs).

## Mitotic index assay

For dose-response experiments, RPE-1 cells were trypsinized 48 h post-transfection and transfected with the siRNA a second time. Immediately afterwards, 4500 cells/well were seeded in 96-well, black, optically clear flat-bottom plates (CellCarrier, PerkinElmer) in triplicate and then treated with the inhibitor for 48 h at the indicated concentrations. Reversine was used at 500 nM. Cells were then fixed and immunofluorescence performed as previously described [84]. For NALM-6 knock-out cell lines, 20,000 cells/well were seeded in triplicate in 96-well, black, optically clear flat-bottom plates that were first coated with fibronectin (12 μg/mL final concentration; Sigma-Aldrich) for 1 h at room temperature and washed once with PBS. Cells were then treated for 48 h with the inhibitor at the indicated concentrations. Fixation and immunofluorescence were carried out as for RPE-1 cells.

Images for the mitotic index analysis in NALM-6 and RPE-1 cells were acquired on an Opera CHKN/QEHS ver. 2.0.1.14101 (Perkin Elmer) microscope using a 20X air immersion objective. DAPI and 488 nm signals were acquired for 20 fields/well. Using the Acapella 2.6 software, the find nuclei method B was used to select the nuclear population to be analyzed. The mitotic index corresponds to the percentage of pHH3 positive cells (i.e., 488 nm fluorescence) relative to the total number of DAPI-stained nuclei. Experiments were performed in triplicate.

## Microscopy and image analysis

Live imaging was performed using a spinning-disk confocal system (Yokogawa CSU-X1 5000) mounted on a fluorescence microscope (Zeiss Axio Observer Z1) using an Axiocam 506 mono camera (Zeiss), 40X oil objective (NA 1.4) and ZEN software (Zen). hTERT RPE-1 cells expressing H2B-GFP and α-Tub-mRFP were transfected with siRNAs and treated with BI2536 or DMSO. Fourty-eight hours later, cells were trypsinized, reseeded and transfected a second time, still in the presence of BI2536 or DMSO. For filming, cells were plated on Lab-Tek II chambered coverglass (Thermo Fisher Scientific), and maintained at 37°C with 5% $CO_2$. Cells were filmed 72 h after the first transfection and treatment with BI2536 or DMSO. For Figs 5A, 7 and S2A, fixed cells were imaged using an LSM 700 laser scanning confocal microscope (Zeiss) with a 40X oil objective (NA 1.4) and ZEN software. For Fig 6B and 6D, fixed cells were imaged using an LSM 880 laser scanning confocal microscope (Zeiss) with a 40X oil objective (NA 1.3) using ZEN software. For measurements of mitotic spindle length and centromere dispersion, image analysis was performed with Zen software (Zeiss) as previously described [85]. Measurements of nuclear solidity and circularity were carried out using the analyze particles tool in ImageJ. Images were first processed for Z projections and export using a customized script in Fiji. CENP-A total intensity was calculated by multiplying the area and the mean intensity obtained with Zen software after tracing the outline of the DAPI-stained nuclei.

## Cellular fractionation

After 72 h of treatment with BI2536 (or DMSO), RPE-1 cells were harvested, centrifuged at 2000 rpm for 5 min, washed in cold PBS containing protease inhibitors and centrifuge again. Cell pellets were lysed in lysis buffer (50 mM Tris-HCl pH 7.5, 150 mM NaCl, 1% triton, 10% glycerol, PMSF, aprotinin and leupeptin). Lysates were incubated on a wheel at 4°C for 15 min, passed 5 times through a needle (27G x 1/2) using a syringe and incubated another 20 min on a wheel at 4°C. Complete lysis was verified under the microscope. Extracts were then centrifuged at 4°C for 10 min at 21 000 g and soluble fractions were collected. The remaining pellets were then washed with lysis buffer containing only 0.2% triton and centrifuged again at 21 000 g for 5 min. Supernatants were discarded and pellets (crude chromatin fractions) were resuspended in 1/10 volume (relative to total lysate) of lysis buffer containing 0.2% triton. For Western blots, aliquots of both fractions were added an equal volume of 2x Laemmli buffer (Sigma-Aldrich) and heated at 95°C for 5 min.

## Statistics

Most statistical analyses were done using GraphPad Prism. The significance of GO term [86,87] enrichments was calculated using Fisher's exact test as implemented in R and subsequently corrected for multiple testing using FDR. To assess the significance of the number of interactions in the protein interaction network, we first removed hit proteins with at least 200 interaction partners in BioGRID and/or IntAct (updated March 2020; ELAV1, LARP7, HEXIM1, CAND1, YWAE1, PDHA1, TUBB, SPDL1, PRKAR1A, RPS27A, RPS23, MAPRE1, DYNLT1), counted the number of unique interactions between different proteins in the list of hits [71] and then performed the following degree-preserving shuffling procedure 10,000 times. For each interaction partner (proteome-wide) of each protein in the list, we selected a random protein from the network with proportional probability to its interaction degree (i.e., number of partners). For each of the 10,000 simulated networks, we counted the number of times the randomly selected protein was itself a hit, which was always lower than 71, corresponding to an estimated p-value of less than 0.0001.

## Supporting information

**S1 Fig. Validation of selected PLK1 inhibition enhancers.** NALM-6 cells were infected with two different sgRNA constructs per gene and with 2 control sgRNAs. After selection with antibiotics, cells were treated with the indicated drugs at the $IC_{30}$ concentrations or with DMSO for 96 hours. Percentages of cell proliferation in the presence of the drug relative to the DMSO control for each cell line are shown. Error bars: range of values from 2 independent experiments. **A.** BI2536 8.1 nM. **B.** BI6727 12.5 nM.**C.** GSK461364A 8.4 nM. **D**. All results in parallel, showing correlation. Dashed lines: % proliferation of the 2 controls. Coordinate values used to generate graphs are available in S8 Data.
(PDF)

**S2 Fig. *PRR14L* is an enhancer of PLK1 inhibition in cytokinesis. A.** Example images of phenotypes observed by immunofluorescence in RPE-1 cells depleted of PRR14L using siRNA and treated with 5 nM BI2536 ($IC_{30}$). **B.** Validation of PRR14L siRNA efficiency. **C.** Quantification of binucleated and multinucleated cells observed after the indicated treatments. Values are averages of 4 experiments in which >350 cells were scored per conditions in each experiment. ** $p < 0.01$ in Student unpaired T test. ns: non-significant. Coordinate values used to generate graphs are available in S9 Data.
(PDF)

**S3 Fig. Analysis of NALM-6 cell lines obtained after CRISPR targeting of *KIF18A* and *SKA1*. A.** Positions targeted by the sgRNAs relative to known domains and motifs in the primary structure of KIF18A and SKA1. **B.** Indel distributions of NALM-6 cells selected for the expression of the indicated sgRNAs along with Cas9. **C.** Summary of data from NALM-6 CRISPR cell lines analyzed in B. Total % $R^2$: percentage of the allelic population that could be reliably analyzed. **D.** Western blot analysis for KIF18A expression in the indicated cell lines. Coordinate values used to generate graphs are available in S10 Data.
(PDF)

**S4 Fig. Inactivation of KIF18A or SKA1 sensitizes cells to mitotic arrest induced by PLK1 inhibition using GSK461364A. A.** Targeting *KIF18A* or *SKA1* by CRISPR sensitizes NALM-6 cells to the GSK461364A-induced mitotic arrest. For A and B, the mitotic index was measured after immunofluorescence for pHH3. Representative experiments are shown. Data points are averages of triplicates ±SD. **B.** Silencing KIF18A or SKA1 by siRNA sensitizes hTERT RPE-1 cells to the GSK461364A-induced mitotic arrest. Coordinate values used to generate graphs are available in S11 Data.
(PDF)

**S5 Fig. Validation of siRNA depletions in RPE-1 cells. A.** Relative levels of *SKA1* and *KIF18A* mRNA were quantified by RT-qPCR. Two independent experiments of siRNA depletions were done in duplicates, resulting in 4 values. Averages ±SD are shown. For each gene (*SKA1* or *KIF18A*), values were normalized by setting the si-Ctrl + DMSO average to 1 (C: calibrator). BI2536 was added at the $IC_{30}$ concentration (5 nM). **B.** Western blots showing the levels of KIF18A protein in all transfections. Coordinate values used to generate graphs are available in S12 Data.
(PDF)

**S1 Table. Summary of CRISPR screens conducted with 3 PLK1 inhibitors at different concentrations.** The concentrations of compounds used, their effects on cell proliferation obtained during the screens and the numbers of enhancer and suppressor genes obtained with

very high confidence (False Discovery Rate; FDR < 0.05) are indicated.
(XLSX)

**S2 Table. Ranking of genetic modifiers of partial PLK1 inhibition identified in the screen.**
For the 3 Plk1 inhibitors used, RANKS score, p-values and FDR are presented for each gene
monitored in the screen. Values obtained from combining all 3 screens are shown in the first
columns and were used to rank enhancer genes (negative mean scores) and suppressor genes
(positive mean scores). Combined p-values were calculated using Fisher's method.
(XLSX)

**S3 Table. Oligonucleotides used in RT-qPCR in this study.** Oligos used in RT-qPCR to mea-
sure the mRNA levels of the indicated genes. Reference of the detected isoform(s) according to
NCBI, Universal Probe Library number used and the efficiency of each pair of oligos following
a standard curve for each gene tested are included (see Materials & Methods).
(PDF)

**S1 Video. RPE-1 cell expressing H2B-GFP and α-Tub-mRFP and treated with Non-target
siRNA (si-Ctrl) and DMSO only (complement to Fig 4).** Anaphase onset occurs with a nor-
mal timing. Images were acquired every 5.5 min on a spinning-disc confocal microscope. The
time of NEBD was set as time zero. Speed: 3 frames per second.
(AVI)

**S2 Video. RPE-1 cell expressing H2B-GFP and α-Tub-mRFP and treated with non-target
siRNA (si-Ctrl) and BI2536 at $IC_{30}$ (complement to Fig 4).** Anaphase onset occurs with a
normal timing. Images were acquired every 5.5 min on a spinning-disc confocal microscope.
The time of NEBD was set as time zero. Speed: 3 frames per second.
(AVI)

**S3 Video. RPE-1 cell expressing H2B-GFP and α-Tub-mRFP and treated with siRNA
against KIF18A and DMSO only (complement to Fig 4).** Anaphase onset is delayed. Images
were acquired every 5.5 min on a spinning-disc confocal microscope. The time of NEBD was
set as time zero. Speed: 3 frames per second.
(AVI)

**S4 Video. RPE-1 cell expressing H2B-GFP and α-Tub-mRFP and treated with siRNA
against KIF18A and BI2536 at $IC_{30}$ (complement to Fig 4).** Anaphase does not occur.
Images were acquired every 5.5 min on a spinning-disc confocal microscope. The time of
NEBD was set as time zero. Speed: 3 frames per second.
(AVI)

**S5 Video. RPE-1 cell expressing H2B-GFP and α-Tub-mRFP and treated with siRNA
against SKA1 and DMSO only (complement to Fig 4).** Anaphase onset is delayed. Images
were acquired every 5.5 min on a spinning-disc confocal microscope. The time of NEBD was
set as time zero. Speed: 3 frames per second.
(AVI)

**S6 Video. RPE-1 cell expressing H2B-GFP and α-Tub-mRFP and treated with siRNA
against SKA1 and BI2536 at $IC_{30}$ (complement to Fig 4).** Anaphase does not occur. Images
were acquired every 5.5 min on a spinning-disc confocal microscope. The time of NEBD was
set as time zero. Speed: 3 frames per second.
(AVI)

**S7 Video. Micronucleation does not occur after anaphase in a cell where PLK1 is partially inhibited (complement to Fig 6).** RPE-1 cell expressing H2B-GFP and α-Tub-mRFP and treated with non-target siRNA (si-Ctrl) and BI2536 at $IC_{30}$. Images were acquired every 5.5 min on a spinning-disc confocal microscope. The time prior to anaphase onset was set as time zero. Speed: 3 frames per second.
(AVI)

**S8 Video. Micronucleation occurs after anaphase in a cell where KIF18A is depleted and PLK1 is partially inhibited (complement to Fig 6).** RPE-1 cell expressing H2B-GFP and α-Tub-mRFP and treated with siRNA against KIF18A and BI2536 at $IC_{30}$. Images were acquired every 5.5 min on a spinning-disc confocal microscope. The time prior to anaphase onset was set as time zero. Speed: 3 frames per second.
(AVI)

**S1 Data. Fig 1E Numerical Data.**
(XLSX)

**S2 Data. Fig 2 Numerical Data.**
(XLSX)

**S3 Data. Fig 3 Numerical Data.**
(XLSX)

**S4 Data. Fig 4 Numerical Data.**
(XLSX)

**S5 Data. Fig 5 Numerical Data.**
(XLSX)

**S6 Data. Fig 6 Numerical Data.**
(XLSX)

**S7 Data. Fig 7 Numerical Data.**
(XLSX)

**S8 Data. S1 Fig Numerical Data.**
(XLSX)

**S9 Data. S2 Fig Numerical Data.**
(XLSX)

**S10 Data. S3 Fig Numerical Data.**
(XLSX)

**S11 Data. S4 Fig Numerical Data.**
(XLSX)

**S12 Data. S5 Fig Numerical Data.**
(XLSX)

## Acknowledgments

We are grateful to Helder Maiato for the hTERT RPE-1 cells WT and expressing H2B-GFP and α-Tub-mRFP. We also thank Vincent Poupart for creating scripts to process large numbers of images, Karine Audette at the IRIC high-throughput screening platform for help with image acquisition and analysis using Opera and Christian Charbonneau at the IRIC

microscopy platform for generous help and advice. We thank members of the Archambault lab for useful discussions.

## Author Contributions

**Conceptualization:** Karine Normandin, Jasmin Coulombe-Huntington, Thierry Bertomeu, Mike Tyers, Vincent Archambault.

**Data curation:** Jasmin Coulombe-Huntington.

**Formal analysis:** Karine Normandin, Jasmin Coulombe-Huntington, Corinne St-Denis, Alexandre Bernard, Mohammed Bourouh, Thierry Bertomeu, Vincent Archambault.

**Funding acquisition:** Mike Tyers, Vincent Archambault.

**Investigation:** Karine Normandin, Jasmin Coulombe-Huntington, Corinne St-Denis, Alexandre Bernard, Mohammed Bourouh, Thierry Bertomeu, Vincent Archambault.

**Methodology:** Karine Normandin, Jasmin Coulombe-Huntington, Corinne St-Denis, Alexandre Bernard, Mohammed Bourouh, Thierry Bertomeu, Mike Tyers, Vincent Archambault.

**Project administration:** Mike Tyers, Vincent Archambault.

**Supervision:** Karine Normandin, Thierry Bertomeu, Mike Tyers, Vincent Archambault.

**Writing – original draft:** Karine Normandin, Jasmin Coulombe-Huntington, Thierry Bertomeu, Mike Tyers, Vincent Archambault.

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
