## [Decision Letter · Decision Letter 0]

2 Jun 2023

Dear Dr Archambault,

Thank you very much for submitting your Research Article entitled 'Genetic enhancers of partial PLK1 inhibition reveal hypersensitivity to kinetochore perturbations' to PLOS Genetics.

The manuscript was fully evaluated at the editorial level and by independent peer reviewers. The reviewers appreciated the attention to an important topic but identified some concerns that we ask you address in a revised manuscript.

We therefore ask you to modify the manuscript according to the review recommendations. Your revisions should address the specific points made by each reviewer.

Yours sincerely,

Michael Snyder, Ph.D.

Academic Editor

PLOS Genetics

David Kwiatkowski

Section Editor

PLOS Genetics

Reviewer's Responses to Questions

**Comments to the Authors:**

Reviewer #1: In this manuscript, the authors report results from a genome-wide CRISPR screen designed to identify enhancers and suppressors of PLK1 inhibitor-dependent suppression of cell proliferation. The screen utilized three different PLK1 inhibitors and identified a number of modifiers involved in cell cycle regulation, kinetochore assembly and function, and the microtubule cytoskeleton. A number of the strongest enhancers of PLK1 have known roles in kinetochore function or regulation of kinetochore microtubule dynamics, suggesting that the kinetochore functions of PLK1 may be particularly dependent on high PLK1 activity. The authors chose two of the identified enhancers, SKA1 and KIF18A, to investigate further in NALM-6 and RPE-1 cells. Combining PLK1 inhibitors at IC30 with siRNAs targeting SKA1 or KIF18A resulted in severe chromosome alignment and mitotic arrest defects, as well as nuclear shape abnormalities. While the mechanistic basis of the enhancer effects from SKA1 and KIF18A knockdown were not completely addressed, the data presented are convincing and provide an important step towards identifying combinatorial, anti-mitotic treatments worthy of additional exploration. A few minor concerns are provided below for the authors to consider.

1. Since PLK1 has a known role in CENP-A recruitment, it could be informative to determine if the severity of mitotic phenotypes scales with CENP-A levels at centromeres.

2. In Figure 5, Ska1 siRNA cells appear to be multipolar. Were these included in the centromere distribution analyses? If so, it should be explained how these results were compared to those from bipolar spindles and how they might affect the dispersion ratios presented.

3. The authors may want to consider avoiding the term “nuclear reassembly defects” to describe the nuclear shape abnormalities reported in Figure 6. “Nuclear reassembly” suggests the process of nuclear envelope formation after mitosis. It is not clear from the data presented that the integrity of the nuclear envelope was tested, and I think the authors are likely referring to defects in forming a single nucleus around all chromosomes.

Reviewer #2: In the manuscript by Normandin, et. al., the authors probe the genome for proteins whose depletion is synthetically interactive with Polo Like Kinase 1(PLK1) inhibition. They identify numerous interactors involved in the cell cycle, unsurprisingly. Further evaluation leads the authors to conclude that combination therapies between PLK1 and proteins involved in chromosome segregation, mitotic spindle function, and nuclear reformation in G1, among others, would be a promising strategy for clinical exploration.

PLK1 is an important cell cycle kinase that has been targeted for cancer therapeutics. PLK1 has long been known to regulate numerous mitotic events including but not limited to centrosome maturation, chromosome condensation, kinetochore function, cytokinesis, and various aspects of mitotic exit including centromere epigenetic regulation. Possibly due to these pleiotropies, PLK1 inhibitors have seen limited success in drug discovery. Here the authors identify the mitotic kinesin KLP18a, a plus end directed family 8 kinesin, as a potential target for combinatorial therapies.

Overall this is a well conducted study that probes a possibly critical question in therapeutics, identification of cocktail type treatments that result in better targeting of cancerous cell types. I am generally in favor of accepting this paper, however I have some reservations that dampen my enthusiasm.

1) The main concern I have is the lack of new insight into mitotic mechanisms. I left this paper feeling like I could have predicted every interactor discovered based on previous publications. This is likely not the main point of the paper, however still wish I had learned more about mitosis from the work. In looking at the figures (5A for instance) kinetochores appear to have lost “integrity” as discrete loci. I find this very interesting, among other phenotypes, and hope the authors follow on with further, more basic and mechanistic studies in the future.

2) There have been few cell level retrospectives on how cancer drugs such as Taxol/Paclitaxol kill cells. Mitchison has conducted a series of studies on Taxol treatment at clinically relevant doses and found that, if I may oversimplify, catastrophic exit from mitosis is the predominant form of death. This seems similar to the observation here of nuclear reformation etc. It might be nice to have some discussion of this point.

3) Along those lines, I find the most important part of this is not the cell biology (which is not so insightful), rather the clinical implications. More of an observation than anything else.

4) Finally, the centromere dispersion assay is not very informative in this context. Many spindle/kinetochore perturbations produce essentially the same thing, increased dispersion of centromeres on the spindle. Live cell imaging to measure congression would be a more appropriate assay.

**Have all data underlying the figures and results presented in the manuscript been provided?**

Reviewer #1: Yes

Reviewer #2: Yes

PLOS authors have the option to publish the peer review history of their article (what does this mean?). If published, this will include your full peer review and any attached files.

Reviewer #1: No

Reviewer #2: No

---

## [Editor Report · Decision Letter 1]

6 Aug 2023

Dear Dr Archambault:

We are pleased to inform you that your manuscript entitled "Genetic enhancers of partial PLK1 inhibition reveal hypersensitivity to kinetochore perturbations" has been editorially accepted for publication in PLOS Genetics. Congratulations!

Yours sincerely,

Michael Snyder, Ph.D.

Academic Editor

PLOS Genetics

David Kwiatkowski

Section Editor

PLOS Genetics

Comments from the reviewers (if applicable):

**Data Deposition**

http://datadryad.org/submit?journalID=pgenetics&manu=PGENETICS-D-23-00388R1

**Press Queries**

---

## [Editor Report · Acceptance letter]

20 Aug 2023

PGENETICS-D-23-00388R1 

Genetic enhancers of partial PLK1 inhibition reveal hypersensitivity to kinetochore perturbations 

Dear Dr Archambault, 

We are pleased to inform you that your manuscript entitled "Genetic enhancers of partial PLK1 inhibition reveal hypersensitivity to kinetochore perturbations" has been formally accepted for publication in PLOS Genetics! Your manuscript is now with our production department and you will be notified of the publication date in due course.

With kind regards,

Lilla Horvath

PLOS Genetics

On behalf of:
